# Mapping the architecture of the initiating phosphoglycosyl transferase from *S. enterica* O-antigen biosynthesis in a liponanoparticle

Greg J Dodge[1], Alyssa J Anderson[1], Yi He[2], Weijing Liu[2], Rosa Viner[2], Barbara Imperiali[1]*

[1]Department of Biology and Department of Chemistry, Massachusetts Institute of Technology, Cambridge, United States; [2]Thermo Fisher Scientific, San Jose, United States

*For correspondence: imper@mit.edu

**Abstract** Bacterial cell surface glycoconjugates are critical for cell survival and for interactions between bacteria and their hosts. Consequently, the pathways responsible for their biosynthesis have untapped potential as therapeutic targets. The localization of many glycoconjugate biosynthesis enzymes to the membrane represents a significant challenge for expressing, purifying, and characterizing these enzymes. Here, we leverage cutting-edge detergent-free methods to stabilize, purify, and structurally characterize WbaP, a phosphoglycosyl transferase (PGT) from the *Salmonella enterica* (LT2) O-antigen biosynthesis. From a functional perspective, these studies establish WbaP as a homodimer, reveal the structural elements responsible for dimerization, shed light on the regulatory role of a domain of unknown function embedded within WbaP, and identify conserved structural motifs between PGTs and functionally unrelated UDP-sugar dehydratases. From a technological perspective, the strategy developed here is generalizable and provides a toolkit for studying other classes of small membrane proteins embedded in liponanoparticles beyond PGTs.

## eLife assessment

This **valuable** manuscript provides **solid** methodologies for utilizing SMALP nanodisks for oligomer characterization. The authors present a platform for capturing and studying native membrane protein oligomerization and subsequent cryoEM analysis. The specific application of the method to WbaP, a membrane-bound phosphoglycosyl transferase, adds to our understanding of glycoconjugate production in bacteria. This manuscript would be of interest to those focusing on native membrane protein studies and antimicrobial resistance.

## Introduction

Glycoconjugates are complex heterogeneous biopolymers found on the cell surface and secreted from all cells. In prokaryotes, (*Tytgat and Lebeer, 2014*) glycoconjugates play important roles in survival, (*Chen et al., 2004*) antibiotic resistance, (*Campos et al., 2004*) immune evasion and immunogenicity (*Tan et al., 2020*; *Bentley et al., 2006*), and biofilm formation (*Vu et al., 2009*) Genetic disruption or inhibition of the enzymes involved in glycoconjugate assembly can result in deleterious effects in many microorganisms and have been reported to attenuate virulence and pathogenicity (*Chua et al., 2021*; *Hong and Reeves, 2016*; *Yethon et al., 2000*) Despite the overwhelming compositional diversity of prokaryotic glycoconjugates, the underlying logic for their biosynthesis is often

**Figure 1.** Biosynthesis of O-antigen repeat units in *S.enterica* serovar typhimurium. The pathway is initiated by the transfer of a phospho-Gal onto undecaprenol phosphate (UndP), catalyzed by the LgPGT WbaP. A series of glycosyltransferases (GTs) add additional sugars to the nascent repeat unit (RU). The repeat unit is then flipped across the membrane by a Wzx-class flippase, and RUs are polymerized through the coordinated action of the Wzy polymerase and Wzz chain-length regulatory protein.

conserved and most commonly localized to cell membranes (*Whitfield et al., 2020a*). Thus, structural and mechanistic studies of the machinery for glycoconjugate assembly are of considerable interest although the membrane localization of glycoconjugate assembly enzymes poses a significant hurdle for in-depth characterization.

The logic of the Wzx/Wzy-dependent glycoconjugate biosynthesis pathways is highly conserved and comprises a series of enzyme-catalyzed transfer reactions using nucleoside diphosphate sugar donor substrates and a membrane resident polyprenol phosphate (PrenP) acceptor such as undeca-prenol phosphate (UndP). These pathways result in the production of a wide array of glycoconjugates (*Islam and Lam, 2014*). O-antigen biosynthesis in *Salmonella enterica* serovar Typhimurium LT2 is a well-studied example of a Wzx/Wzy-dependent pathway (*Jiang et al., 1991*; *Samuel and Reeves, 2003*; *Kalynych et al., 2014*). This pathway utilizes a PGT known as WbaP (formerly RfbP) to catalyze the transfer of phospho-galactose from UDP-galactose to UndP, forming Und-PP-galactose (*Wang et al., 1996*; *Wang and Reeves, 1994*; *Patel et al., 2012*; *Patel et al., 2010*; *Saldías et al., 2008*). This reaction represents the initial membrane-committed step in O-antigen repeat unit (RU) biosynthesis and is followed by a series of glycosyl transfer reactions to form the O-antigen RU. The Und-PP-linked tetrasaccharide RU is then flipped across the inner bacterial membrane, polymerized into full-length O-antigen, and appended to the lipid A core for display on the outer membrane (*Figure 1*; *Whitfield et al., 2020a*; *Islam and Lam, 2014*; *Whitfield et al., 2020b*; *Woodward et al., 2010*; *Liu et al., 1996*).

PGTs belong to one of two superfamilies based on the membrane topology of the catalytic core structure (*Figure 2, Table 1*). Polytopic PGTs (polyPGTs), such as MraY, (*Chung et al., 2013*; *Oluwole et al., 2022*) have a catalytic domain comprising multiple transmembrane helices (TMHs). while the catalytic core of monotopic PGTs (monoPGTs) contains only a single reentrant membrane helix (RMH) (*O'Toole et al., 2021a*). Analysis of the monoPGT superfamily utilizing a sequence similarity network (SSN) recently classified several family members each featuring variation around the core catalytic domain (*O'Toole et al., 2021b*). Small monoPGTs (Sm-PGTs), such as PglC from the *Campylobacter* protein glycosylation pathways only include the catalytic core (*Lukose et al., 2015*; *Ray et al., 2018*). Bifunctional monoPGTs (Bi-PGTs) include functional domains fused to either the N- or C-terminus of the monoPGT catalytic domain. Finally, large monoPGTs (Lg-PGTs), which are the most abundant, include both a predicted transmembrane four-helix bundle and a domain of unknown function (DUF) N-terminal to the monoPGT core catalytic domain. The DUF is highly conserved (PF13727) and has been computationally annotated as a CoA-binding domain, however, the roles of the conserved accessory domains in the Lg-PGTs remain unclear. The *S. enterica* WbaP is a prototypic Lg-PGT that has been biochemically characterized (*Wang et al., 1996*; *Wang and Reeves, 1994*; *Patel et al., 2012*; *Patel et al., 2010*; *Saldías et al., 2008*; *Dodge et al., 2023*). Initial computational assignment of the

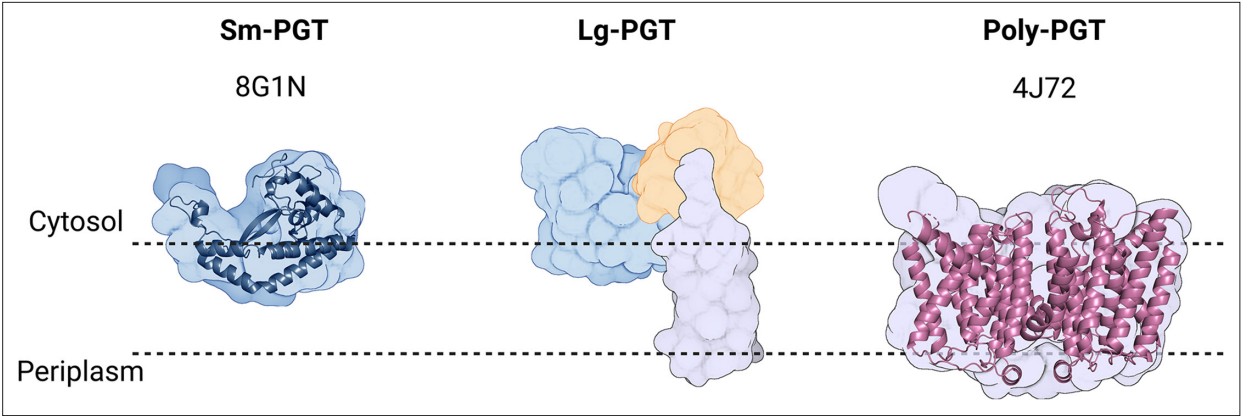

**Figure 2.** Overview of common phosphoglycosyl transferase (PGT) family members. Sm-monoPGTs do occupy one leaflet of the membrane, comprise only the catalytic core of monotopic PGTs (monoPGTs) (blue surface), and are exemplified by the structurally characterized PglC from *Campylobacter concisus* (dark blue cartoon). Large monoPGTs (Lg-PGTs) feature the C-terminal conserved catalytic core domain (blue surface) and two uncharacterized N-terminal domains. These domains feature a predicted transmembrane helix bundle (purple surface) and a domain of unknown function (gold surface). Poly-PGTs such as MraY from *Aquifex aeolicus* (magenta cartoon) comprise a catalytic domain, which is structurally distinct Sm- or Lg-PGTs and have been demonstrated to dimerize in their active form.

topology of Lg-PGTs misassigned the topology of the RMH as a TMH, confounding interpretations of biological data, as well as the subcellular localization of the DUF in these proteins (*James et al., 2013*; *Furlong et al., 2015*; *Entova et al., 2018*). Currently, although the structure of PglC, a sm-PGT from *Campylobacter concisus* has been determined by X-ray crystallography, (*Ray et al., 2018*) there is no experimental structure determination of any Lg-PGT to provide insight into the roles of the auxiliary domains or their interactions with the catalytic core. Furthermore, PglC was purified using detergent, potentially obscuring interactions that depend on the native membrane-bound environment of PglC.

Amphiphilic PrenPs such as UndP and decaprenol phosphate are central to glycoconjugate assembly in bacteria and represent an essential cellular resource with no other known function (*Barreteau et al., 2009*). The low abundance of UndP (ca. 0.1% of membrane lipids) (*Barreteau et al., 2009*; *Entova et al., 2019*) suggests that PrenP-dependent pathways must be subject to regulation to ensure that UndP supplies are maintained for critical pathways. For example, sequestration of UndP by disruption of O-antigen production in *E. coli* results in gross morphological defects and cell lysis (*Jorgenson and Young, 2016*). Despite this, our understanding of the regulation of Wzx/Wzy-dependent glyco-conjugate biosynthesis remains limited. PGTs are likely to be subject to regulation, as they act as the initial 'gatekeeper' enzyme to transfer glycosyl phosphates from soluble NDP-sugar substrates onto the membrane-embedded UndP. NDP-sugars used by PGTs are often highly modified or are shared with fundamental cellular processes such as glucose metabolism, placing a further metabolic burden on organisms to modulate PGT activity. Recently, the sm-PGT CapM from capsular polysaccharide biosynthesis in *Staphylococcus aureus* was demonstrated to be activated via phosphorylation by a

**Table 1.** Summary of the differences between phosphoglycosyl transferase (PGT) superfamilies.

|  | Sm-PGT | Lg-PGT | Poly-PGT |
|---|---|---|---|
| Catalytic Domain Topology | Monotopic | Monotopic | Polytopic |
| Domains | PGT only | PGT & uncharacterized accessory domains | PGT only |
| Evolutionary Conservation | Prokaryotic | Prokaryotic | Prokaryotic & eukaryotic |
| Oligomeric State | Monomer *Ray et al., 2018*; *Anderson et al., 2023* | Unknown | Dimer *Chung et al., 2013*; *Oluwole et al., 2022* |
| monoPGT SSN Abundance in superfamily *O'Toole et al., 2021b* | 38% | 47% | N/A |

regulatory kinase (*Rausch et al., 2019*). Studies of a WbaP ortholog from *Streptococcus pneumoniae* suggest that residues in the DUF may modulate PGT function or overall pathway flux, raising the possibility that the accessory domains found in Lg-PGT may serve a regulatory function (*James et al., 2013*; *Cartee et al., 2005*).

We have recently reported a robust pipeline for solubilizing and purifying Lg-PGTs from several organisms, including WbaP from *S. enterica* (*Dodge et al., 2023*). The strategy includes in vivo cleavage of an N-terminal SUMO tag via the protease Ulp1, application of the amphiphilic polymer SMALP-200 (formerly SMA30) for direct solubilization from *E. coli* membranes into styrene-maleic acid liponanoparticles (SMALPs), and purification via a dual-strep tag, which is exposed upon cleavage by Ulp1 in cellulo. The resulting protein is highly pure and stabilized in a native-like membrane environment by the SMALP. WbaP assembled in SMALP was vitrified on grids for CryoEM screening, and an initial dataset yielded 2D classes with unexpected symmetry. As these experiments were the first to study purified Lg-PGTs in a native-like lipid bilayer, the putative symmetry observed in SMALP prompted the investigation of the oligomeric state of Lg-PGTs in the liponanoparticles.

Here, we build upon the successful membrane protein solubilization techniques to further investigate the architecture of *S. enterica* WbaP in SMALP. We leverage AlphaFold modeling, crosslinking experiments, mass spectrometry, and CryoEM to map the structure of *S. enterica* WbaP in a native-like lipid bilayer environment. Insights from these experiments establish the oligomeric state of *S. enterica* WbaP as a dimer, facilitate the production of a soluble truncation construct to probe the role of the DUF, and highlight evolutionary relationships between Lg-PGTs and other enzymes from glycoconjugate biosynthetic pathways. These findings demonstrate SMALP as a useful platform for capturing and studying native oligomerization state and for CryoEM of small membrane proteins. These structural studies now allow us to develop a clearer picture of the initial steps of glycoconjugate biosynthesis and membrane positioning of Lg-PGTs. Finally, we assign a nucleotide-sensing regulatory function to the DUF, establishing a role for this highly conserved non-catalytic domain in Lg-PGTs.

## Results
### Production of Lg-PGTs in Styrene maleic acid liponanoparticles (SMALP)
### Characterization of *S. enterica* WbaP in SMALP

We took a multipronged approach to investigate the putative oligomerization state of *S. enterica* in liponanoparticles (*Figure 3*, *Figure 3—figure supplement 1A*). As SMALPs may display high heterogeneity we turned to size-exclusion chromatography (SEC) to remove aggregated material or free SMA polymer that had carried over through initial purification. A small peak was observed corresponding to the void volume of the column, but most of the material eluted in a sharp peak with a retention time of 11.09 min corresponding to a molecular weight of 464 kDa (*Figure 3—figure supplement 1B*). The predicted molecular weight of the expression construct of WbaP is 61.2 kDa. However, as the SMA and lipid accompanying WbaP in the SMALP may influence separation on SEC relative to globular standards, we sought alternative methods to accurately determine the mass of the WbaP SMALPs in solution. Size exclusion chromatography with multi-angle light scattering (SEC-MALS), mass photometry, and Direct Mass Technology mode (Orbitrap-enabled charge detection mass spectrometry) (*Kafader et al., 2019*) were utilized to assess the molar mass of WbaP in SMALP. In agreement with the initial SEC experiments, WbaP eluted in a sharp peak on an analytical SEC column in line with MALS (*Figure 3A*). The polydispersity of this peak was calculated to 1.001, and the molar mass was calculated to 255±9 kDa. Accurate determination of the mass of a species analyzed by SEC-MALS requires a precise assignment of the refractive index increment (dn/dc) of each component of the analyte, however, the refractive index increment (dn/dc) for SMA polymers is under debate in the literature (*Hesketh et al., 2020*; *Thomsen, 2020*) Despite the low polydispersity and narrow mass range assigned to this peak, the three-component nature (protein, lipid, and polymer) of WbaP-embedded SMALPs prevents deconvolution of the mass of the protein component. To this end, we turned to Direct Mass Technology mode and mass photometry, techniques that can accurately determine the mass of macromolecules in solution regardless of their composition (*Kafader et al., 2019*; *Olerinyova et al., 2021*) When analyzed by mass photometry or Direct Mass Technology mode, WbaP-containing SMALPs were determined to have a molecular weight of ~240 kDa (*Figure 3B and C*). As the monomer molecular weight of the WbaP construct was calculated to 61.2 kDa, and the

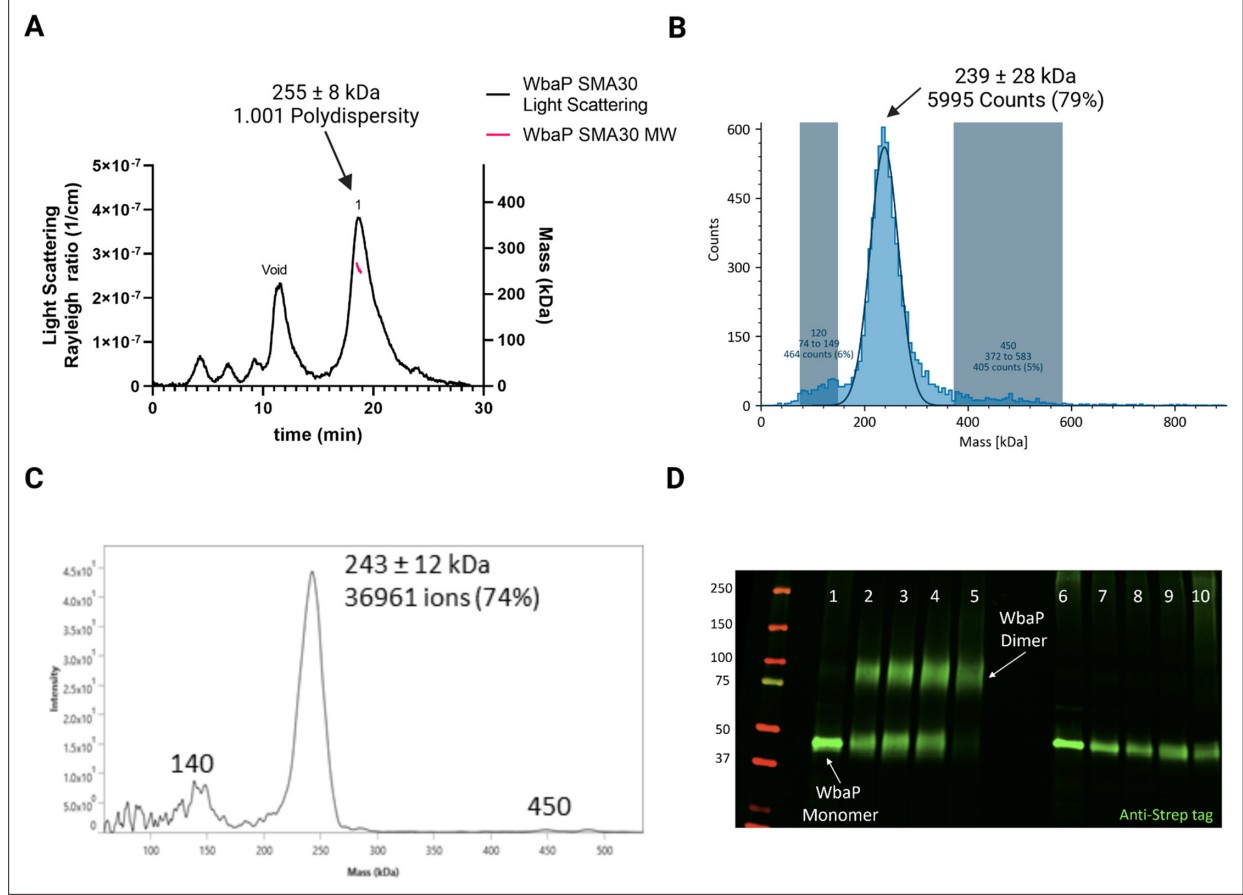

**Figure 3.** Characterization of *S.enterica* WbaP in SMALP. (**A**) WbaP in styrene-maleic acid liponanoparticles (SMALP) analyzed by size exclusion chromatography with multi-angle light scattering (SEC-MALS). A small peak is observed at the void volume of the column, while the main peak has a calculated molecular weight of 255 ± 8 kDa, with a polydispersity value of 1.001. (**B**) WbaP in SMALP analyzed by mass photometry. The sample is monodisperse, and the main species has an apparent molecular weight of 239 ± 28 kDa. (**C**) Mass spectrum of WbaP in SMALP analyzed by Direct Mass Technology mode. (**D**) Western blot analysis of Lysine-reactive dithiobis(succinimidyl propionate) (DSP) crosslinker reacted with WbaP in SMALP. Lanes: 1: Control, 2: 0.1 mM DSP, 3: 0.25 mM DSP, 4: 1 mM DSP, 5: 5 mM DSP, 7: 0.1 mM DSP reduced, 8: 0.25 mM DSP reduced, 9: 1 mM DSP reduced, 10: 5 mM DSP reduced.

The online version of this article includes the following source data and figure supplement(s) for figure 3:

**Source data 1.** Raw data for the blot shown in *Figure 3D*.

**Figure supplement 1.** Purification and characterization of *S.enterica* WbaP in styrene-maleic acid liponanoparticle (SMALP).

**Figure supplement 1—source data 1.** Raw SDS-PAGE gel data relating to *Figure 3—figure supplement 1A*.

**Figure supplement 2.** Crosslinking strategy for *S.enterica* WbaP in styrene-maleic acid liponanoparticle (SMALP).

SMA along with encapsulated lipids can add a variable amount of mass depending on the solubilized protein, we reasoned that WbaP could exist as a dimer or trimer in SMALP.

To further define the oligomeric state of WbaP in SMALP, we utilized amine-reactive crosslinking probes. By selecting crosslinkers of a defined length, we reasoned that we could covalently trap oligomers and distinguish dimer from trimer using non-reducing sodium dodecyl-sulfate polyacrylamide gel electrophoresis (SDS-PAGE) (*Figure 3—figure supplement 2A*). A small panel of commonly used crosslinkers was selected to screen against WbaP in SMALP (*Figure 3—figure supplement 2B*). Of these, the crosslinker dithiobis(succinimidylpropionate) (DSP), with a spacer length of 12.0 Å, readily crosslinked WbaP (*Figure 2D*, *Supplementary file 1A*). Crosslinking efficiency was dependent on DSP concentration, with complete crosslinking observed with 5 mM DSP after a 5 min reaction. Crosslinking

could be reversed by the addition of a Dithiothreitol (DTT). Fully crosslinked WbaP migrated at ~2 x the molecular weight of the non-crosslinked species, confirming a WbaP dimer in SMALP.

## CryoEM of *S. enterica* WbaP in SMALP

With high confidence in the monodispersity of our sample and the oligomeric state of *S. enterica* WbaP in SMALP, we continued with CryoEM structure analysis. Prior to this work, only a small screening dataset had been collected on a Talos Arctica instrument (*Dodge et al., 2023*) Grids were prepared using the highly purified eluent from SEC, movies were collected using a Titan Krios G3i microscope. 2D classes from extracted particles closely matched those observed earlier, and initial *ab-initio* 3D models displayed features consistent with a liponanoparticle, as well as two distinct protein lobes outside of the membrane. However, various refinement strategies failed to yield even moderate-resolution reconstructions. Reasoning that the auto-generated refinement mask may include SMA and lipid regions, we sought to generate a reasonable model to assist in manual masking during data processing.

## Creation and validation of *S. enterica* WbaP dimer model

We turned to AlphaFold to generate models suitable for docking and refinement (*Jumper et al., 2021*) Although the prediction of *S. enterica* WbaP available from the AlphaFold protein structure database has high confidence throughout most of the model (*Figure 4A*, *Figure 4—figure supplement 1A, B*), we noted a region of moderate confidence between the DUF and the PGT catalytic domain. In particular, the placement of a predicted β-hairpin motif from Asn251 to Gln268 stood out as anomalous, as this motif was flipped away from the remainder of the protein, approximately 25 Å from the DUF, and ~37 Å from the PGT domain (*Figure 4A*). As the AlphaFold multimer routine has been reported to successfully predict oligomeric complexes, (*Evans et al., 2021*; *Mirdita et al., 2022*) we assessed whether a dimer model of *S. enterica* WbaP would represent a better starting point for docking and refinement into the CryoEM reconstruction. The dimer was also predicted with high overall confidence (*Figure 4B*, *Figure 4—figure supplement 1A*). Unexpectedly, the β-hairpin motif that had poor placement in the monomer model was predicted to mediate domain swapping within the dimer model, (*Bennett et al., 1994*; *Liu and Eisenberg, 2002*) with the β-hairpin from chain A continuing a β-sheet structure present in the DUF of chain B (*Figure 4B and C*). The predicted local distance difference test (pLDDT) score assigned to each amino acid by AlphaFold can be used to judge the quality of a predicted model on a per-residue basis (*Jumper et al., 2021*; *Evans et al., 2021*) Amino acids with pLDDT scores >70 are considered to have high confidence (*Akdel et al., 2022*) For the β-hairpin domain swap region, the average pLDDT score is 78, indicating high confidence in the placement of this motif (*Figure 4—figure supplement 1A*). Significantly, the prediction of structures of other Lg-PGTs reveals that this β-hairpin domain swap within a homodimer is highly conserved (*Figure 4—figure supplement 2*).

Based on the earlier success with crosslinking *S. enterica* WbaP in SMALP, we further investigated the predicted domain swapping using a targeted crosslinking strategy. A set of Cys-pair variants of WbaP were constructed such that thiol-reactive crosslinkers with limited crosslinking length could be used to 'staple' the WbaP dimer at the predicted domain swap interface (*Figure 5A–C*). For these experiments, we employed two different crosslinking strategies. For the first Cys-crosslinking experiment, we utilized Cu phenanthroline to catalyze the direct oxidation of Cys residues to form a disulfide bond between Cys residues within 2.05 Å. For the second strategy, we utilized dibromobimane (bBBr) (*Kim and Raines, 1995*) to form a covalent crosslink between Cys residues within 4.88 Å. We observed concentration-dependent crosslinking for two discrete sets of crosslinking pairs along the predicted domain swap interface using either Cu phenanthroline or bBBr (*Figure 5D and E*). No crosslinking was observed for wild-type protein using either crosslinking strategy, demonstrating that there are no native cysteines within crosslinking distance, (*Figure 5E*) as predicted by the AlphaFold dimer model.

As a final validation of the predicted dimer model, we used crosslinking mass spectrometry (XLMS). Both DSS and tBu-PhoX were chosen for crosslinking, as DSS is commonly used for XLMS, (*Chavez and Bruce, 2019*; *Piersimoni et al., 2022*) and tBu-PhoX has membrane-penetrating properties (*Figure 6A*; *Jiang et al., 2022*). After crosslinking, WbaP was exchanged into DDM micelles (*Hesketh et al., 2020*), and subjected to protease digestion. In-solution digestion resulted in fragments covering 93.86% *S. enterica* WbaP, and 49 unique crosslinking sites were identified (*Figure 6B*,

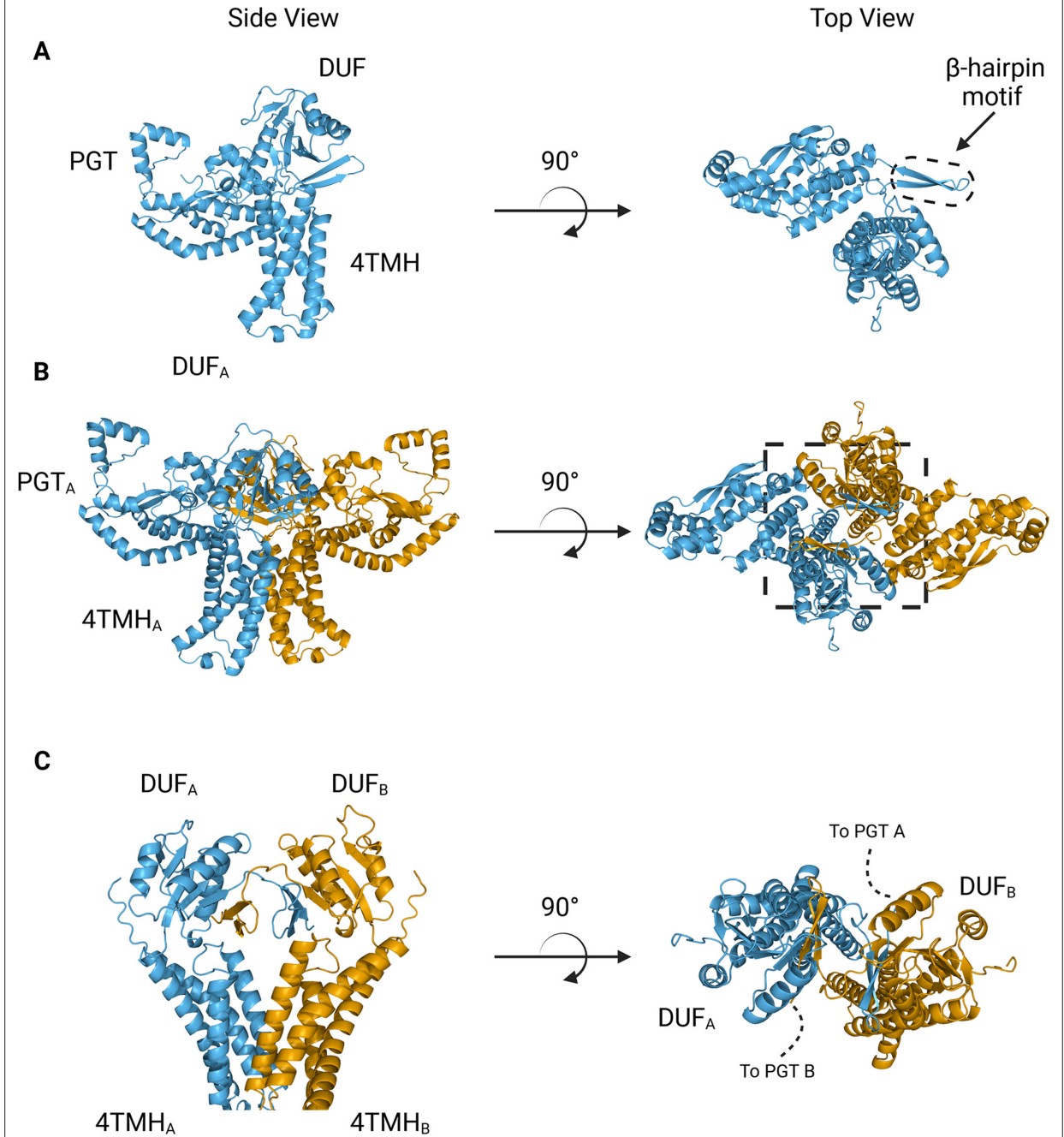

**Figure 4.** AlphaFold predictions of *S.enterica* WbaP monomer and dimer. (**A**) WbaP monomer prediction. (**B**) WbaP dimer prediction. (**C**) Close-up view of predicted dimer interface showing the interdigitating β-hairpin motifs between $DUF_A$ and $DUF_B$. Phosphoglycosyl transferase (PGT) domains hidden for clarity.

The online version of this article includes the following figure supplement(s) for figure 4:

**Figure supplement 1.** Analysis of *S.enterica* WbaP AlphaFold prediction.

**Figure supplement 2.** AlphaFold dimer predictions for various large monoPGTs (Lg-PGTs), along with predicted local distance difference test (pLDDT) plots and predicted aligned error (PAE) plots.

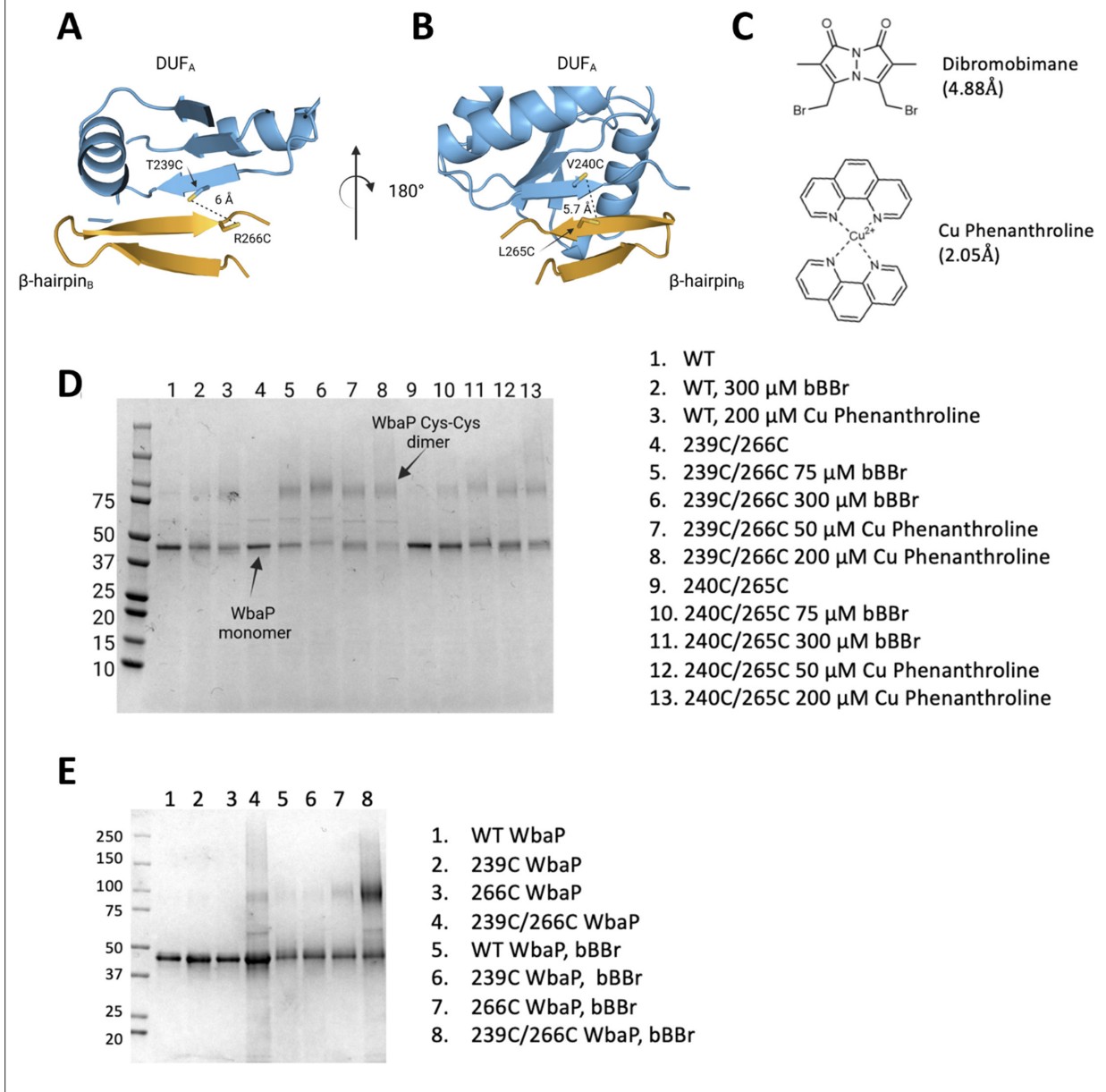

**Figure 5.** Probing the predicted interface of the WbaP dimer. (**A**, **B**) AlphaFold prediction with Cys pair variants modeled as sticks. The β-hairpin of one monomer is shown in sky blue and the continuing β- strand of the other monomer is shown in orange. (**C**) Structure and crosslinking radii of thiol-reactive crosslinkers. (**D**) Crosslinking of Cys pair WbaP variants in styrene-maleic acid liponanoparticle (SMALP) using dibromobimane (bBBr) and Cu Phenanthroline. (**E**) bBBr crosslinking of single and double WbaP Cys variants in SMALP.

The online version of this article includes the following source data for figure 5:

**Source data 1.** Raw gel data for *Figure 5D*.

**Source data 2.** Raw gel data for *Figure 5E*.

*Figure 6—figure supplement 1*). The majority of the DSS crosslinks localized to the solvent-accessible areas of the PGT domain, while several of the tBu-PhoX crosslinks localized to the membrane-adjacent regions at the putative dimer interface (*Figure 6C*). Inter vs intra-molecular crosslinks were assigned by XMAS (*Lagerwaard et al., 2022*) based on distance constraints for each crosslinker (*Figure 6A*). The Cα−Cα distances bridged by either crosslinker should not exceed 35 Å. The observed inter-chain

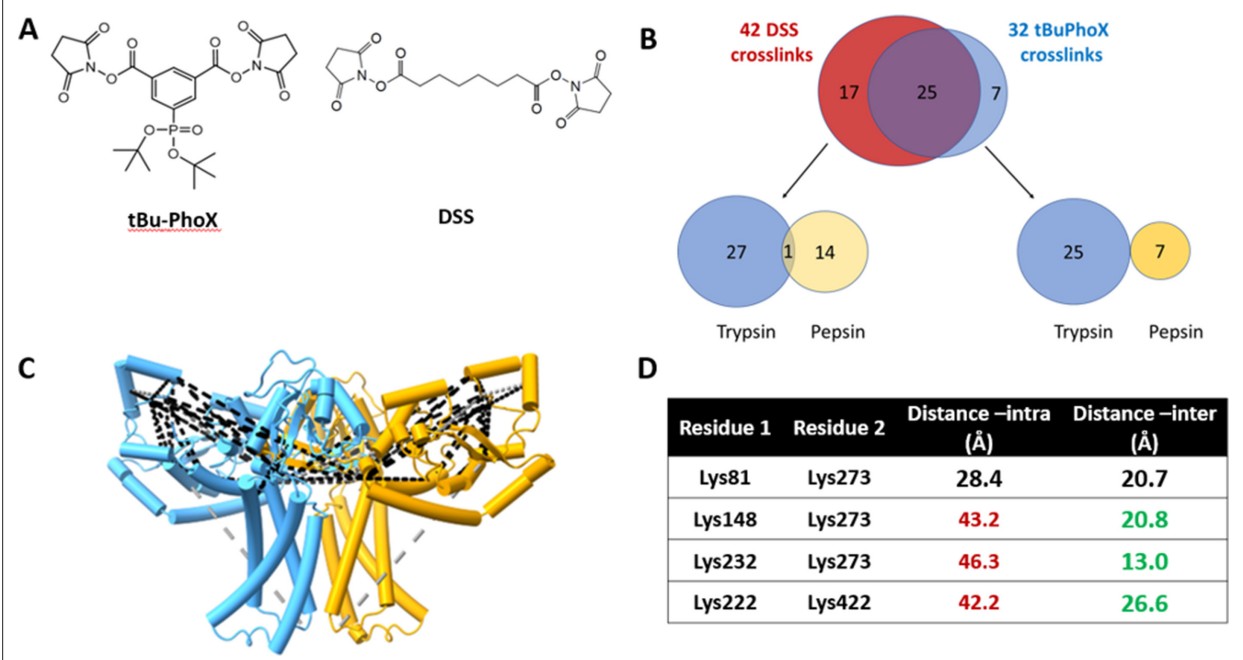

**Figure 6.** Crosslinking mass spectrometry (XLMS) of WbaP in styrene-maleic acid liponanoparticle (SMALP). (**A**) Structures of amino reactive crosslinkers used for XLMS analysis. (**B**) Overview of identified WbaP crosslinks from different protease treatments and crosslinkers. (**C**) Identified crosslinks mapped onto *S. enterica* WbaP AlphaFold dimer model. Disuccinimidyl suberate (DSS) crosslinks are shown in black, tert-butyl disuccinimidyl phenyl phosphonate (tBu-PhoX) crosslinks are shown in gray. (**D**) Residues linked by intermolecular crosslinks. The distance between these residues within a single monomer and between monomers is shown.

The online version of this article includes the following figure supplement(s) for figure 6:

**Figure supplement 1.** Crosslinking mass spectrometry analysis of *S. enterica* WbaP in styrene-maleic acid liponanoparticle (SMALP).

crosslinks agreed with the predicted dimer interface, with Lys273 proximal to Lys146, Lys148, and Lys81. In addition, Lys81 and Lys451 were within crosslinking range (*Figure 6C*). With targeted cross-linking experiments validating the predicted domain swapping and XLMS validating the dimeric oligo-merization state, we proceeded with the AlphaFold dimer model for CryoEM data processing and refinement.

## *S. enterica* WbaP structure

A soft mask was generated using the AlphaFold dimer model of WbaP to mask around phospho-lipid and SMA density in the best unmasked CryoEM reconstruction. Subsequent masked refinement and cleaning of particle stacks led to a moderate-resolution (~4.1 Å by GSFSC) reconstruction from 196,663 particles which was consistent with the AlphaFold model (*Figure 7A and B*, *Figure 7—figure supplement 1*, *Figure 7—figure supplement 2*). Docking and real-space refinement led to the place-ment of 796 out of 1056 residues of the dimer (*Figure 7A and B*). Superposition between the refined model and the AlphaFold dimer prediction results in an RMSD of 2.83 Å, demonstrating close agree-ment between the two models (*Figure 7C*).

The overall architecture of the WbaP dimer is T-shaped, with the two four-TMH bundles forming the central stalk, each capped by a DUF. The dimer interface comprises ~780 Å² of buried surface area, involving 7.1% of the residues of the protein. A modest (176 Å²) interface is found along the four TMH bundles, with the C-terminal half of helix 3 (Ile99 – Phe107) mediating contacts. While the C-ter-minal regions of helix 3 are within interacting distance, the N-termini (Pro82) are ~32 Å apart, creating a large void in the center of the dimer, which is presumably occupied by phospholipid. The remainder of the dimer interface occurs at the DUF domains, mediated almost entirely by the β-hairpin domain crossover discussed above. Based on previous experiments establishing the topology of monoPGT catalytic domains, the DUF domains cap the cytosolic loops between the four TMH bundles (*Ray et al., 2018*; *Furlong et al., 2015*; *Entova et al., 2018*).

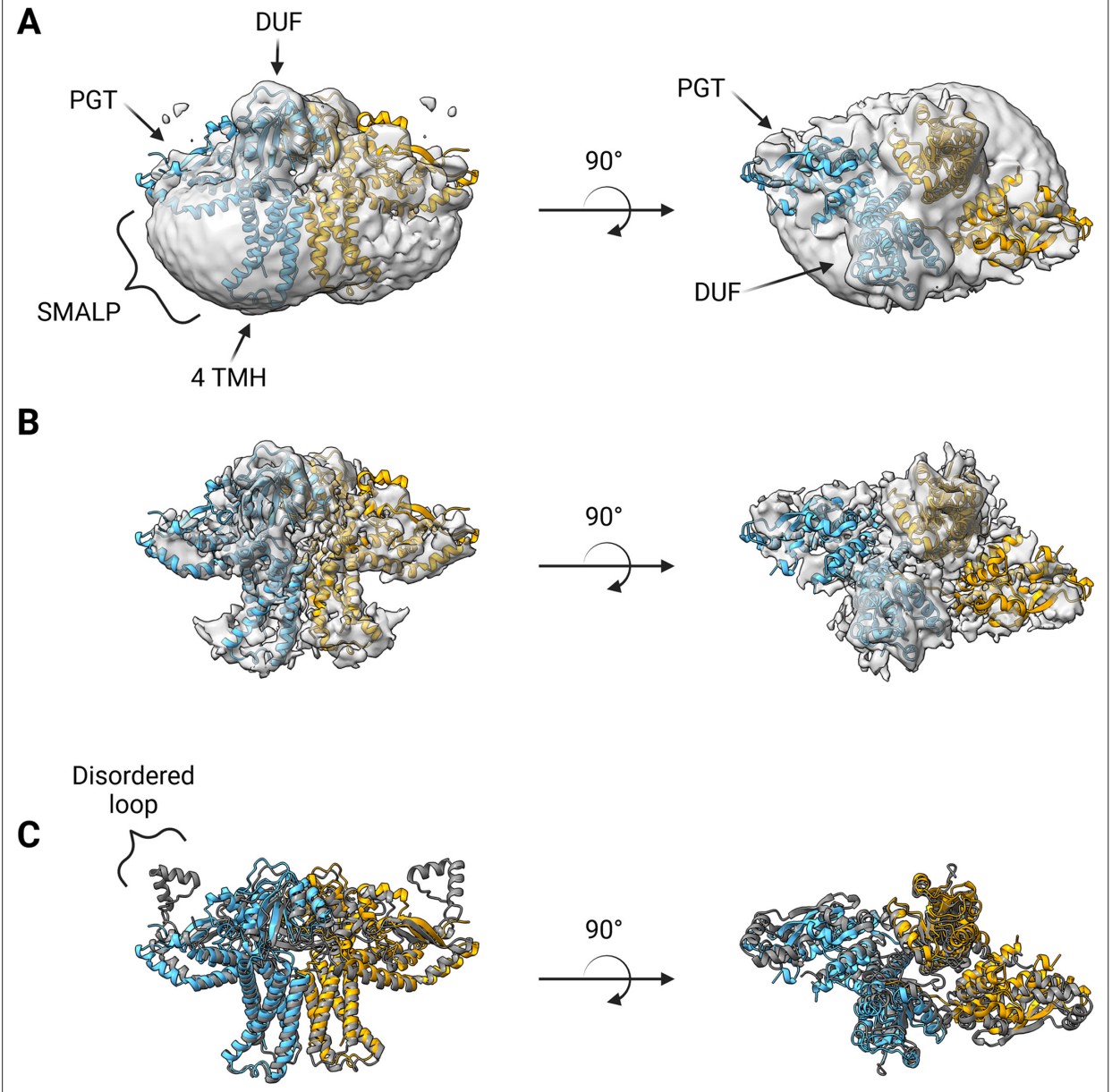

**Figure 7.** Structure of *S.enterica* WbaP in styrene-maleic acid liponanoparticle (SMALP). WbaP dimer colored in blue and orange (**A**) Unsharpened map displaying clear density for stabilizing liponanoparticle. Unsharpened density colored light gray. (**B**) Sharpened map after masked local refinement. Sharpened density colored light gray. (**C**) Superposition of refined WbaP model and full-length AlphaFold prediction, RMSD: 2.83 Å. AlphaFold dimer colored gray. A predicted helix-turn-helix motif lacks density in the experimental map.

The online version of this article includes the following figure supplement(s) for figure 7:

**Figure supplement 1.** EM processing workflow.

**Figure supplement 2.** Local resolution estimation of masked, un-sharpened WbaP reconstruction.

The catalytic domains of the Lg-PGT protrude away from the dimer interface, placing the two active sites 77 Å apart. Despite the large distance between catalytic sites, the interdigitating β-hairpin motif at the dimer interface between the DUF domains places Ser259, found at the apex of the turn of the β-hairpin, only ~23 Å from the active site of the opposite chain. Although the β-hairpin is poorly ordered in the experimental map, the targeted Cys-pair crosslinking confirms the placement

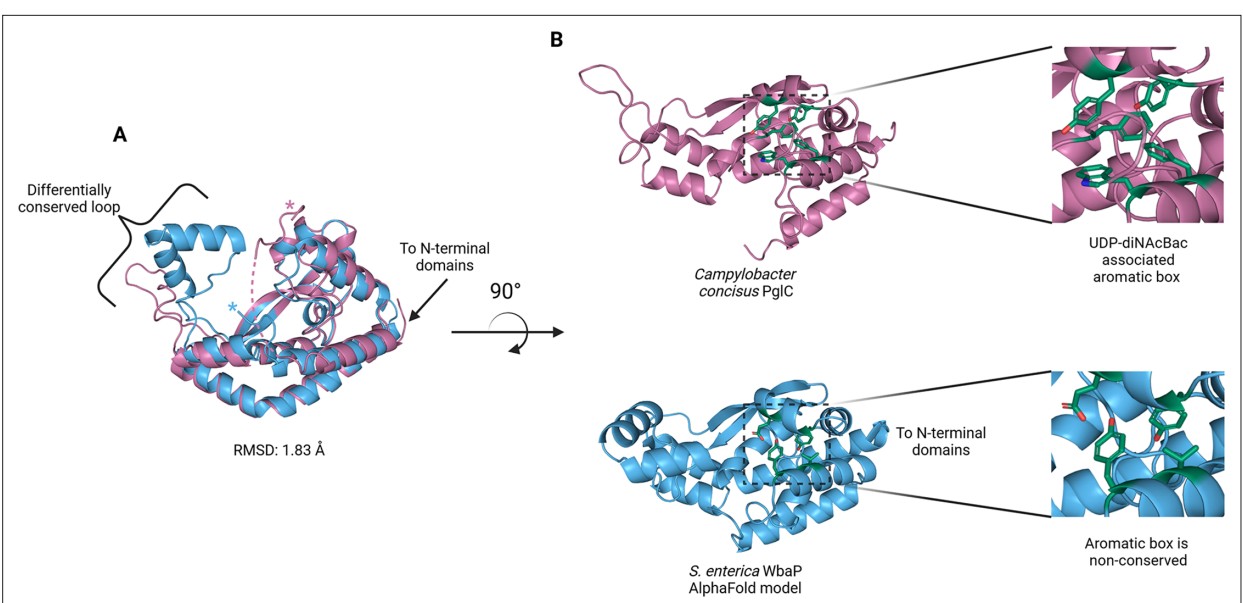

**Figure 8.** Comparison of *S. enterica* WbaP phosphoglycosyl transferase (PGT) domain and PglC from Campylobacter concisus (PDBid 8G1N). WbaP is shown in light blue, PglC is in magenta. (**A**) Superposition of WbaP and PglC. While the overall RMSD is 1.83 Å, regions in PglC which can be used to computationally assign substrate are significantly different in WbaP. A loop region in PglC is replaced with a helix-turn-helix motif in WbaP, and the C-terminus of WbaP is significantly shorter than that of PglC. (**B**) Top: an aromatic box motif in PglC is conserved among PGTs that utilize UDP-diNAcBac. Aromatic box residues are shown as dark green sticks. Bottom: Residues found in *S. enterica* WbaP at homologous positions to aromatic box residues in PglC are shown as dark green sticks. Aromatic box residues are non-conserved in WbaP.

of this motif found in the AlphaFold model. A predicted helix-turn-helix motif from residues 337–369 of the catalytic domain is also disordered in the experimental map (*Figure 7C*). This region is analogous to a mobile loop found in the *C. concisus* PglC, which has been implicated in the formation of the substrate binding pocket (*Anderson et al., 2023*) Although both the mobile loop in PglC and the helix-turn-helix found in WbaP are positioned proximal to the active site, the length, sequence conservation, and structural motifs of this region differ greatly between these two PGTs (*Figure 8*). These modifications around the conserved catalytic site may provide a structural rationale for the differing UDP-sugar substrate specificities between PglC and WbaP. XLMS experiments demonstrate that lysines within the predicted helix-turn-helix of WbaP can crosslink with lysines across the soluble core of the catalytic domain (*Figure 6C*), giving high confidence to the placement of this region in the AlphaFold model, despite the lack of density in the experimental map.

During the course of this work, a structural description of a WbaP ortholog from *E. coli* (NCBI:taxid562) by CryoEM was published (*Uchański et al., 2021*). This protein was purified using both detergent and amphipols, and was complexed with a nanobody fusion to facilitate particle identification and alignment during 3D-reconstruction. Although the maps and models associated with this data are not yet publicly available, visual comparison between *S. enterica* WbaP and published images of the *E. coli* WbaP reveals an overall similar architecture, including disorder in the helix-turn-helix region. However, the handedness of the *E. coli* WbaP reconstruction is inverted compared to *S. enterica* WbaP as well as the predicted AlphaFold model. Notably, AlphaFold multimer was not available at the time of publication of the *E. coli* structure and the handedness may be ambiguous at the observed resolution without a model for guidance.

With a clear idea of the overall architecture of WbaP, we sought to better understand the role of the DUF beyond the stabilization of the dimer interface. To this end, we used both the CryoEM reconstruction and AlphaFold structural models to guide the generation of truncated constructs of WbaP encoding only the DUF. A construct encoding residues 146–272 resulted in the production of soluble protein (*Figure 9—figure supplement 1*), which was then used to screen for potential nucleotide ligands by nano differential scanning fluorimetry (nDSF, *Figure 9—figure supplement*

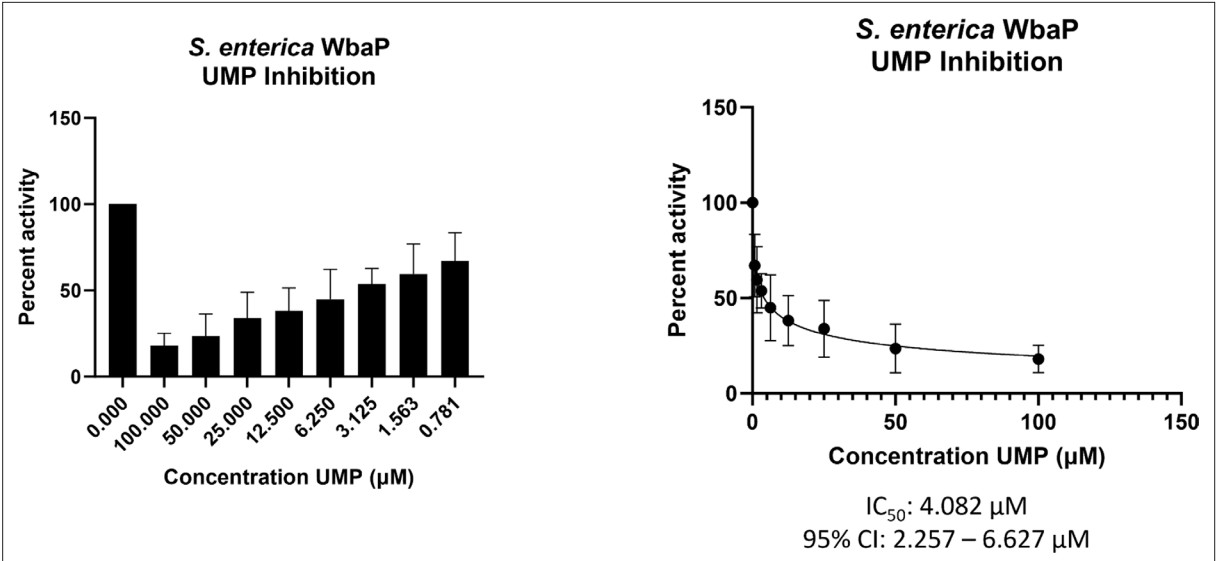

**Figure 9.** WbaP UMP titration assay. Left: Percent activity vs UMP concentration. A UMP-dependent decrease in WbaP activity is observed. Right: Curve-fitting of UMP inhibition data yields an IC50 apparent of 4.1 µM. Data were collected in triplicate with error bars representing standard deviation.

The online version of this article includes the following source data and figure supplement(s) for figure 9:

**Figure supplement 1.** Purification of *S.enterica* WbaP soluble domain of unknown function domain of unknown function (DUF) truncation.

**Figure supplement 1—source data 1.** Raw SDS-PAGE gel data for *Figure 9—figure supplement 1A*.

**Figure supplement 2.** Nano differential scanning fluorimetry (nDSF) of purified soluble WbaP DUF truncation.

*2*). This screen identified UMP, UDP, and UTP as potential stabilizing ligands, and also demonstrated that the UDP-Gal substrate of the catalytic domain destabilized the truncated DUF construct. These results imply that the DUF plays a role in allosteric regulation by sensing nucleotides and nucleotide sugars. Indeed, full-length WbaP was found to be strongly inhibited by UMP, with an apparent $IC_{50}$ of 4.1 µM (*Figure 9*). This is in close agreement with studies on the WbaP ortholog CpsE, for which UMP has an apparent $K_i$ of 3 µM (*Cartee et al., 2005*). The unusual profile of the UMP inhibition curve may reflect UMP binding both competitively to UDP-Gal at the active site, as well as to an allosteric site within the DUF. Thus, it appears that the DUF may act as a regulatory domain, preventing the overuse of limited UndP resources by modulating PGT activity in response to excess UMP production.

## Relationship between Lg-PGTs and PglF-like dehydratases

The N-terminal domains found in WbaP are highly conserved across the Lg-PGT family, (*O'Toole et al., 2021b*) however, a search of the PDB for structural homologs of the domain alone returned only nucleotide-binding domains with limited homology to the DUF. To sample a larger set of structural models, we utilized the recently described Foldseek tool to parse the AlphaFold model database (AFDB proteome) (*van Kempen et al., 2023*). In this analysis, although most of the aligned models were other PGTs, the N-terminal domains of WbaP also exhibited structural homology to models generated from a class of NDP-sugar dehydratases involved in glycoconjugate biosynthesis (*Figure 10*). Homologs identified by Foldseek include PglF from *Campylobacter* and Cap5D from *Staphylococcus*. This family of DHs is well characterized both structurally and mechanistically, however, constructs used for most structural studies are truncated at the N-terminus and do not include the transmembrane domain or DUF technical reasons (*Riegert et al., 2017*; *Olivier et al., 2006*). As these two classes of enzymes are functionally unrelated, the conservation of their N-terminal domains is puzzling, and warrants re-examination of full-length constructs of PglF-like DHs.

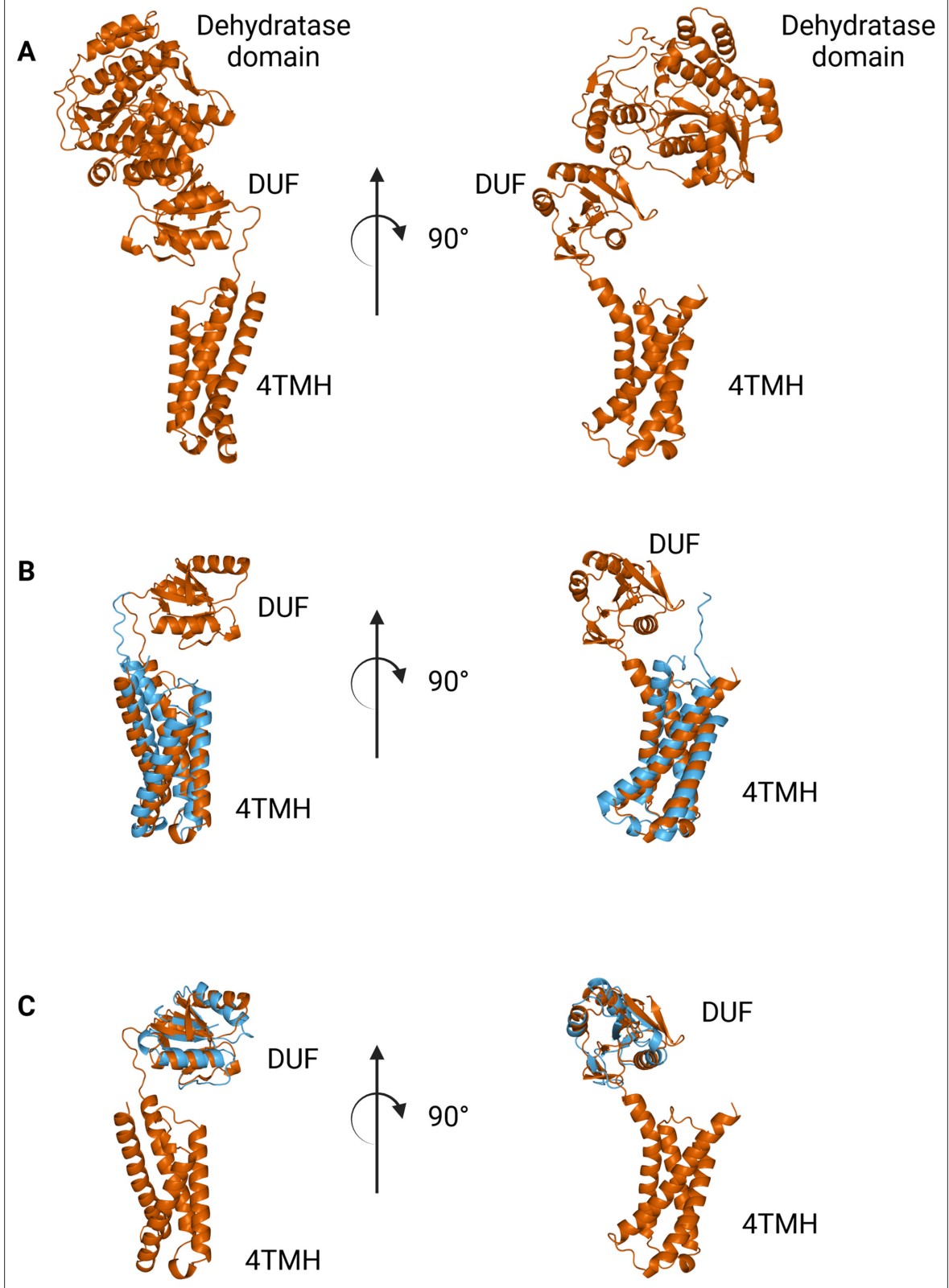

**Figure 10.** Comparison between N-terminal domains from *C.jejuni* PglF (Uniprot: Q0P9D4), a nucleotide sugar dehydratase, and *S. enterica* WbaP. (**A**) AlphaFold prediction of full-length PglF. (**B**) Superposition of *C. jejuni* PglF and *S. enterica* WbaP 4TMH domain. RMSD: 4.3 Å. Dehydratase domain is omitted for clarity. (**C**) Superposition of *C. jejuni* PglF and *S. enterica* WbaP domain of unknown function (DUF) domain. RMSD: 2.3 Å. Dehydratase domain is omitted for clarity.

*Figure 10 continued on next page*

## Discussion

Detailed studies of monoPGTs are hampered by the perennial problems associated with the expression, purification, and structural characterization of membrane proteins. However, the essential role of monoPGTs in the first membrane-committed step of the biosynthesis of diverse classes of prokaryotic glycoconjugates makes structural and functional analysis of this prokaryote-specific enzyme superfamily imperative. Here, we build on our recent advances in the development of an optimized solubilization and purification strategy for the abundant Lg-PGT members of the monoPGT superfamily and present the structural characterization of WbaP, the initiating PGT from O-antigen biosynthesis in *S. enterica*. We anticipate that the strategy applied here will be broadly applicable to other bacterial membrane proteins and provide an efficient route to map the structure of target proteins in a near-native environment, even in the absence of high-resolution structural data. In addition, these methods will be well suited to the study of membrane protein complexes or membrane-bound metabolons, (*Møller, 2010*; *Bassard and Laursen, 2019*) a long-standing goal in the study of glycoconjugate biosynthesis.

The structure presented represents the first monotopic PGT characterized in the native-like membrane of a liponanoparticle. We show that in contrast to the Sm-PGT PglC from *Campylobacter*, which is functional as a monomer, the *S. enterica* Lg-PGT, WbaP, is a tightly associated dimer, with the majority of inter-chain contacts occurring in the highly conserved but previously uncharacterized DUF. Despite computational assignment as a CoA-binding domain, thermal denaturation binding experiments implicate the DUF in uridine nucleotide and UDP-sugar binding. Biochemical assays show that UMP is a potent inhibitor of full-length WbaP activity (*Figure 9*). The profile of the inhibition curve suggests additional UMP interactions beyond the active site in the full-length protein. Taken together, these results point toward the DUF serving both a *structural and functional* role in stabilizing the dimer and allosterically regulating the catalytic domain upon UMP binding. As UMP is produced by PGTs regardless of the sugar moiety of the NDP-sugar substrate, allosteric regulation of PGT catalysis via the DUF represents a 'one-size-fits-all' approach to regulating Lg-PGTs with differing NDP-sugar substrates. DUF-based modulation of PGT function provides a consistent explanation for the in vivo observation that *S. pneumoniae* CpsE accumulates suppressor mutations within its DUF in a system where PGT activity is lethal (*James et al., 2013*; *Cartee et al., 2005*).

Intriguingly, we observe a surprising structural homology between the N-terminal domains of Lg-PGTs and PglF-like dehydratases, which are common UDP-sugar modifying enzymes involved in the biogenesis of unusual carbohydrates for glycoconjugate biosynthesis. In this case, it is tempting to hypothesize that the DUFs may play a larger role in the modulation of the flux of glycoconjugate biosynthesis by subjecting both Lg-PGTs and PglF-like dehydratases to negative feedback by pathway-side products.

Although PglC and WbaP are predicted to catalyze similar chemical transformations, these PGT enzymes exhibit different substrate specificity. PglC shows high specificity for UDP-diNAcBac, while WbaP is specific for UDP-Gal. Recently, structural and sequence elements were identified that facilitated the bioinformatic assignment of substrate specificity to Sm-PGTs that use the same UDP-diNAcBac substrate as PglC (*Anderson et al., 2023*). Interestingly, the major structural deviations between PglC and WbaP occur in these regions (*Figure 8*). An extended loop containing a conserved 'GLLP' motif in diNAcBac-utilizing PGTs is replaced with a predicted helix-turn-helix motif in WbaP. This region is poorly resolved in the experimental CryoEM reconstruction of WbaP, however, lysines in this putative structural motif crosslink with the main domain of the WbaP dimer. Given the apparent high degree of motion in the vitrified WbaP particles at this region, and the proximity of the helix-turn-helix motif to the active site, we envision a mechanism in which the active site 'closes' upon the helix-turn-helix motion towards the center of the dimer, and the active site is 'open' when the motif moves away, as observed in the AlphaFold model. This resembles structural conformers observed in both crystal structures and molecular dynamics simulations of *C. concisus* PglC (*Majumder et al., 2023*). Another structural element in the diNAcBac-specific PGTs that is predictive of substrate specificity

is an aromatic box motif comprising residues in the solvent-accessible region of the protein and a conserved Phe near the C-terminus of this subset of Sm-PGTs (*Anderson et al., 2023*). In WbaP, the C-terminus of the catalytic domain is ~14 amino acids shorter than in the diNAcBac Sm-PGTs, and the aromatic box motif is absent. These subtle changes surrounding the conserved Asp-Glu catalytic dyad are likely to drive the specificity of WbaP to UDP-Gal. As additional UDP-Gal utilizing PGTs are characterized, we envision the creation of a bioinformatic pipeline to identify the specific residues involved in substrate selection, akin to the method applied to the diNAcBac PGTs.

The liponanoparticle-based approach for purification of *S. enterica* WbaP provides a convenient way to add contrast to vitrified particles in CryoEM by way of additional lipid and SMA mass, similar to the rationale for utilization of nanobody fusion strategies in the study of other small membrane proteins (*Uchański et al., 2021*; *Wu and Rapoport, 2021*). Future efforts to improve the overall resolution of Lg-PGT CryoEM reconstructions may benefit from a combination of these approaches whereby mass from a liponanoparticle facilitates particle identification and picking, while bound nanobody or nanobody complexes/fusions reduce structural motion of the bound Lg-PGT and add distinct features to aid in masking and 3D refinement.

# Materials and methods
## Cloning and expression
### Expression of full-length Lg-PGTs

WbaP from *S. enterica* LT2 (Uniprot: P26406) was purified as previously described (*Dodge et al., 2023*). Briefly, C43 cells harboring pAM174 (*Sjodt et al., 2018*) were transformed with a plasmid encoding *S. enterica* WbaP with an N-terminal SUMO tag – linker – dual-strep tag sequence. Protein was expressed using autoinduction (*Studier, 2005*) in 0.5 L terrific broth (*Tartof, 1987*) supplemented with 150 μg/mL kanamycin and 25 μg/mL chloramphenicol. Cells were incubated at 37 °C until the $OD_{600}$ reached ~1.5, and then the temperature was adjusted to 18 ° C and 1 g solid (L)-arabinose was added to each culture. After 18–20 hr expression, cells were pelleted via centrifugation, pellets were transferred to 1-gallon Ziplock freezer bags, manually spread to a uniform thin layer, and frozen at –80 °C.

### WbaP DUF expression and purification

The region encoding residues 146–272 from *S. enterica* WbaP was ordered as a synthetic gene, and cloned into pMCSG7 (*Eschenfeldt et al., 2009*) using Gibson assembly (*Gibson et al., 2009*). The resulting plasmid was transformed into *E. coli* Bl21 (DE3), and protein was expressed using autoinduction (*Studier, 2005*) in 0.5 L cultures supplemented with 100 μg/mL ampicillin at 37 °C. The temperature was adjusted to 18 °C once the $OD_{600}$ reached 1.5, and cultures were incubated for 18 hr. After expression, cells were harvested via centrifugation, transferred to 1-gallon Ziplock freezer bags, spread to a thin layer, and stored at –80 °C.

### Mutagenesis

*S. enterica* WbaP Cys-pair variants were generated using primers designed in the QuikChange primer tool (Agilent). Successful mutations were confirmed by Sanger sequencing. Forward and reverse primers are shown below.

| Mutant | Primers (5'–3') |
| --- | --- |
| T239C | 5'-GCGAAATGACGGAACTACGCAGACGGAGCGACAGTGATG-3' 5'-CATCACTG TCGCTCCGTCTGCGTAGTTCCGTCATTTCGC-3' |
| V240C | 5'-GCGAAATGACGGAACGCATGTGACGGAGCGACAGTGATG-3' 5'-CATCACTG TCGCTCCGTCACATGCGTTCCGTCATTTCGC-3' |
| L265C | 5'-CAGGTTATTCTGAATGCGGCACAACATCACCTCGTGGGAG-3' 5'-CTCCCACG AGGTGATGTTGTGCCGCATTCAGAATAACCTG-3' |
| R266C | 5'-GGTTATTCTGAATGCAAAGCAACATCACCTCGTGG-3' 5'-CCACGAGGTGAT GTTGCTTTGCATTCAGAATAACC-3' |

## Protein purification

### WbaP in SMALP

WbaP was purified in SMALP200 liponanoparticles as previously described (*Dodge et al., 2023*). Frozen cells were resuspended in buffer A (50 mM HEPES pH 8.0, 300 mM NaCl) at 4 mL per g pellet. Resuspended pellets were supplemented with 2 mM $MgCl_2$, 0.06 mg/mL lysozyme (RPI), and 0.5 mg/mL DNase I (Millipore Sigma) and incubated on ice for 30 min. Cells were disrupted via sonication (2 × 90 s, 50% amplitude, 1 s on 2 s off), and the cell membrane was isolated via differential centrifugation (Ti-45 rotor, 9000 g 45 min, reserve supernatant, 140,000 g 65 min). Isolated membranes were diluted to 50 mg/mL using buffer A (assessed by UV absorbance at 280 nm), flash frozen in liquid $N_2$, and stored at –80 °C. Isolated membranes were thawed on ice, and mixed 1:1 (v/v) with a 2% stock solution of SMALP200 (Polyscope) or buffer A and rotated at room temperature for 1 hr. Soluble liponanoparticles were isolated by centrifugation (Ti45 rotor, 160,000 × g 65 min). After centrifugation, pellets were discarded, and the supernatant was flowed over 1 mL Strep-Tactin XT 4flow resin (IBA Biosciences) pre-equilibrated with buffer A. The flowthrough was re-run over the column bed, and the column was washed with 5 mL buffer A. Protein was eluted using 3 mL Buffer A + 50 mM Biotin (buffer B). Protein-containing fractions were identified by UV-vis (Nanodrop) and pooled. Biotin was removed using 3 x tandem 5 mL HiTrap desalting columns (Cytiva) equilibrated with 25 mM HEPES pH 8.0, 150 mM NaCl (buffer C). Protein purity was assessed via SDS-PAGE, and protein concentration was determined via BCA assay (Pierce). Protein was concentrated to 4 mg/mL, and flash frozen in liquid $N_2$, then stored at 80 °C.

### Soluble WbaP truncation

All purification steps were performed on ice. Frozen cell pellets were resuspended in buffer D (50 mM HEPES pH 7.5, 300 mM NaCl, 20 mM imidazole pH 7.5, 5% glycerol) at 4 mL per g cell pellet. Resuspended pellets were brought to 2 mM $MgCl_2$, 0.06 mg/mL lysozyme (RPI), and 0.5 mg/mL DNase I (Millipore Sigma). Cells were incubated on ice for 30 min. Cells were then disrupted via sonication (2 × 90 s, 50% amplitude, 1 s on 2 s off). Insoluble material was removed via centrifugation (Ti45 rotor, 42,000 RPM 60 min). The supernatant was sterile filtered and loaded onto a 5 mL NiNTA His-Trap column using an Akta FPLC pre-equilibrated with buffer D. Protein was eluted using a linear gradient from 0–100% buffer E (50 mM HEPES pH 7.5, 300 mM NaCl, 400 mM imidazole pH 7.5, 5% glycerol) over five column volumes. Peak fractions were pooled, supplemented with TEV protease at a 1:7 TEV:protein ratio, and dialyzed overnight at 4 °C against 3 L buffer F (25 mM HEPES pH 7.5, 150 mM NaCl, 5% glycerol). Inverse NiNTA purification was used to remove the TEV protease and the cleaved 6x His tag. Flowthrough and low-imidazole wash from inverse NiNTA purification were pooled, concentrated to 5 mL, and injected on a Superdex S200 column pre-equilibrated with buffer F (25 mM HEPES pH 7.5, 150 mM NaCl, 5% glycerol). Peak fractions were analyzed via SDS-PAGE, and fractions containing the *S. enterica* WbaP DUF were pooled, concentrated to 2 mg/mL, flash frozen in liquid $N_2$, and stored at –80 °C.

## Characterization of Lg-PGTs in SMALP

### Size exclusion chromatography

*S. enterica* WbaP in SMALP was analyzed to assess monodispersity via size exclusion chromatography (SEC) using an Enrich S650 column (Biorad) pre-equilibrated with buffer C. A 500 μL sample was injected, and peak fractions were pooled and concentrated to 10 mg/mL.

### Size exclusion chromatography with multi-angle light scattering (SEC-MALS)

A 50 μL aliquot of 0.5 mg/mL *S. enterica* WbaP in SMALP was injected onto a WTC-030 fused silica column pre-equilibrated with HEPES buffered saline pH 7.5 (HBS). Eluate was flowed through a DAWN MALS detector (Wyatt) and an Optilab differential refractive index detector (Wyatt). Data were analyzed using ASTRA software (Wyatt), and the system was pre-calibrated with a BSA standard.

### Mass photometry

The *S. enterica* WbaP in SMALP was diluted to 40 nM in buffer C to a final volume of 50 μL. A CultureWell gasket (Grace Bio-Labs) was attached to a 24 mm × 50 mm glass coverslip (Electron

Microscopy Sciences), and 10 µL buffer C was dispensed into a well to acquire focus. A 10 µL aliquot of 40 nM WbaP solution was added to the buffer well to achieve a final concentration of 20 nM WbaP. Following this a 120 s dataset was collected, and the data was analyzed using DiscoverMP (Refyn Ltd). A standard curve was generated using NativeMark unstained protein ladder (Thermo Fisher Scientific).

## Crosslinking experiments

### Crosslinking panel

SMALPs containing *S. enterica* WbaP were reacted with a panel of lysine-reactive crosslinkers with variable chemistries, solubilities, and spacer lengths. Stock solutions were made of each crosslinker as follows: 50 mM dimethyl adipimidate (DMA) in water, 50 mM dimethyl suberimidate (DMS) in water, 25 mM bis[2-(succinimidyloxycarbonyloxy)ethyl]sulfone (BSOCOES) in dry DMSO, and 10 mM 3,3'-dithiobis(sulfosuccinimidyl propionate) (DTSSP) in water. Crosslinkers were immediately added to 15 µM WbaP in buffer C at a 20–40 fold molar excess. After incubation at room temperature for 30 min, the reaction was quenched by the addition of 1 µL of 1.5 M Tris-HCl, pH 8. Samples were analyzed by Western blot to detect the presence of crosslinked oligomers. Crosslinking efficiency was annotated as high (>30% total protein crosslinked to dimer), moderate (<30% total protein crosslinked to dimer), or none (no protein crosslinked to dimer).

### Dithiobis(succinimidylpropionate) (DSP) crosslinking

DSP was dissolved in dry DMSO to a stock concentration of 25 mM and immediately added to a final concentration of 0.1–5 mM to 20 µL of WbaP in SMALP. After incubation at room temperature for 30 min, the reaction was quenched by the addition of 1 µL of 1.5 M Tris-HCl, pH 8. The sample was then divided into two aliquots and reducing (+ DTT) or nonreducing loading dye was added to each sample. Crosslinking was analyzed by SDS-PAGE followed by anti-Strep Western blot analysis.

### Cys-pair variant crosslinking

A 2.5 mM stock solution of bBBr in 20% acetonitrile was prepared. bBBr was added to 15 µM WbaP to a final concentration of either 75 or 300 µM. After incubation at room temperature for 20 min, loading dye was added and SDS-PAGE was performed to assess the WbaP oligomerization state.

A 10 mM 1,10-phenanthroline DMSO stock and a 5 mM CuSO$_4$ aqueous stock were prepared. CuSO$_4$ and 1,10 phenanthroline were added to 20 µL of 15 µM WbaP to final concentrations of 50 and 100 µM or 200 and 400 µM, respectively. The reactions were allowed to proceed at room temperature for 20 min, followed by the addition of non-reducing loading dye and SDS-PAGE analysis.

## Structural biology experiments

### Electron microscopy details

Data were collected at the MIT Characterization.nano facility. Optimal freezing conditions were screened using the Vitrobot system (Thermo Fisher Scientific). Grids were clipped and screened on a Talos Arctica G2 (Thermo Fisher Scientific) equipped with a Falcon 3EC camera (Thermo Fisher Scientific). Optimal particles were observed in grid GD5-3, containing 2 mg/mL *S. enterica* WbaP. Based on these preliminary results, 4481 movies of SMA30-solubilized *S. enterica* WbaP were collected on a Titan Krios G3i (Thermo Fisher Scientific, *Supplementary file 1B*) equipped with a K3 camera (Gatan, Inc). Motion correction, CTF estimation, particle extraction, *ab-initio* reconstruction, and all refinement steps were conducted using Cryosparc V4 *Punjani, 2020*; *Punjani et al., 2017*.

### AlphaFold modelling

Lg-PGTs from *T. thermophilus* (Uniprot: A0A510HWX9), *A. hydrophila* (Uniprot: B3FN88), *E. coli* (Uniprot: P71241), and *S. pneumoniae* (Uniprot: Q9ZII5) were selected for structure prediction. Alpha-Fold models were generated as described previously *Dodge et al., 2023*. Sequences containing two chains of each Lg-PGT were used as input for a local installation of colabfold *Mirdita et al., 2022*, and predictions were generated using the flags `--amber`, `--templates`, `--num-recycle 3`, `--use-gpu-relax` activated.

## Structure refinement and modeling

PHENIX version 1.20.1 was used to process the AlphaFold prediction, dock, rebuild, and real-space refine the processed model, and validate the resulting structural model of *S. enterica* WbaP (*Adams et al., 2010*; *Afonine et al., 2018*). UCSF ChimeraX was used to visualize CryoEM volumes (*Pettersen et al., 2021*). Structural figures were generated using PyMOL (*Schrodinger, 2015*). The protein topology cartoon was generated using Pro-origami (*Stivala et al., 2011*). The EMBL PISA server was used to analyze the dimer interface (*Krissinel and Henrick, 2007*).

## Mass spectrometry

### Cross-linking, protein digestion, and IMAC enrichment

*S. enterica* WbaP in SMALP nanoparticles was prepared in 20 mM HEPES buffer pH 7.0 at a concentration of 0.6 mg/mL. 5 mM disuccinimidyl suberate (DSS) or 2 mM tert-butyl disuccinimidyl phenyl phosphonate (tBu-PhoX)(Thermo Fisher Scientific) in DMSO was added to the solution and incubated for 1 hr at room temperature. Reactions were quenched with 20 mM of Tris-HCl, pH 8.0, for 15 min. N-dodecyl β-D-maltoside (DDM, 10% stock, Thermo Fisher Scientific) was added to the sample to a final concentration of 1% and the mixture was incubated on ice for 30 min. To remove SMA, MgCl$_2$ (50 mM stock) was added to a final concentration of 4 mM, and the resulting solution was incubated at 4 °C for 1 hr, and then centrifuged at 21,000 × g at 4 °C for 1 hr. The supernatants were transferred to fresh microfuge tubes and diluted 1:1 with 0.1% SDS, 25 mM DTT, and incubated at 50 °C for 1 hr. Chloroacetamide was added to 25 mM and incubated at RT for 30 min in the dark before acetone precipitation overnight at –20 °C. The samples were washed twice with 90% acetone and the pellet was vortexed with 25 mM ammonium bicarbonate until re-solubilized. Enzymatic digestion was carried out with either trypsin in 0.1% Rapigest (Waters) (1:20 ratio) or pepsin (1:50 ratio). The trypsin digestion was stopped after 16 hr with 1% formic acid (FA). Cross-linked peptides were desalted using Pierce peptide desalting spin column (Thermo Fisher Scientific) and dried. The tBu-PhoX cross-linked peptides were enriched as described previously (*Jiang et al., 2022*).

### XL-MS data acquisition

Samples were separated by reverse phase-HPLC using a Thermo Scientific Vanquish Neo system connected to an EASY-Spray PepMap 75 µm x 25 cm column over a 60 min 3–65% gradient (A: water, 0.1% formic acid; B: 80% acetonitrile, 0.1% formic acid) at 300 nL/min flow rate. The crosslinked samples were analyzed on the Orbitrap Eclipse Tribrid mass spectrometer with Instrument Control Software version 4.0. 0. Cross-linked samples were analyzed using an HCD-MS2 acquisition strategy with 30% normalized collision energy (NCE). MS1 and MS2 scans were acquired in the orbitrap with a respective mass resolution of 60,000 and 30,000. MS1 scan range was set to m/z 375–1,400 at 100% AGC target, 118 ms maximum injection time, and 60 s dynamic exclusion. MS2 scans were set to an AGC target of 200%, 70ms injection time, isolation window of 1.2 m/z. Only cross-linked precursors at charged states +3 to+8 were subjected to MS2.

### Direct Mass Technology mode measurements

Single ion measurement using Direct Mass Technology mode was performed on Thermo Scientific Q Exactive UHMR. Prior to Direct Mass Technology mode MS analysis, sample was buffer exchanged into 100 mM ammonium acetate using Zeba spin column (7 kDa MWCO, Thermo Fisher Scientific). Sample was loaded into Au/Pd-coated borosilicate emitters (Thermo Scientific, ES388) for nano-ESI. For Instrument parameters, ion transfer target m/z and detector optimization were set to 'high m/z.' In-source trapping was set at −100 V. Trapping gas pressure readout was ~2 × 10$^{-11}$ mbar. Data were acquired at 200 K @m/z400 resolution setting. Data analysis was done by STORIBoard software (Proteinaceous).

## Data analysis

Spectral raw data files were analyzed using Proteome Discoverer 3.0 software (Thermo Fisher Scientific) with XlinkX node 3.0 using the non-cleavable or non-cleavable fast search algorithms for cross-linked peptides and SEQUEST HT search engine for unmodified peptides and loop-links/mono-links. MS1 ion mass tolerance: 10 ppm; MS2 ion mass tolerance: 20 ppm. Maximal number of missed cleavages:

2; minimum peptide length: 6; max. modifications: 4; peptide mass: 500–8,000 Da. Carbamidomethylation (+57.021 Da) of cysteines was used as a static modification. PhoX or DSS cross-linked mass modifications for lysine, the protein N-terminus, and methionine oxidation (+15.995 Da) were used as variable modifications. Data were searched for cross-links against a protein database generated from protein identifications using WbaP and the *E. coli* proteome retrieved from UniProt as search space. The false discovery rate (FDR) was set to 1% at CSM and cross-link levels. The maximum XlinkX score was set to be greater or equal to 40. Post-processing and visualization were carried out using the XMAS plug-in for ChimeraX (*Lagerwaard et al., 2022*; *Pettersen et al., 2021*).

### Nano differential scanning fluorimetry (nDSF) of truncated *S. enterica* WbaP

Data were collected using a Prometheus Panta nDSF instrument (Nanotemper). An initial panel of 20 nucleotides was chosen to screen for potential DUF ligands (*Supplementary file 1C*). Purified DUF at 1 mg/mL was diluted 1:1 with buffer C supplemented with 400 µM ligand, resulting in a final protein concentration of 0.5 mg/mL and 200 µM ligand. Data were processed using Pr.Analysis software (Nanotemper).

### WbaP inhibition assay

WbaP activity assays were carried out as described previously, (*Dodge et al., 2023*) with the following modifications. Prior to the addition of [³H]-labeled UDP-Gal, reactions were incubated at room temperature with UMP for 5 min. A twofold serial dilution of UMP concentrations from 100 µM to 0.78 µM was utilized to assess UMP inhibition compared to a control reaction containing no additional UMP. Reactions were quenched after 10 min. Data were collected in triplicate (biological replicates), normalized to percent activity of the control, and fit using the [inhibitor] vs normalized response –variable slope function in GraphPad Prism Version 9.5.1 for Windows, GraphPad Software, San Diego California, USA, https://www.graphpad.com/.

### Materials availability

All plasmids described in this work will be made available upon request to the corresponding author.

## Acknowledgements

We thank Ed Brignole and Christopher Borsa (MIT.nano) for their support with electron microscopy and Prof. Karen Allen (Boston University) for their thoughtful comments on the manuscript. This work was funded by NIH grants R01 GM131627 and GM039334 (to BI) and F32 GM134576 (to GJD).

## Additional information

### Funding

| Funder | Grant reference number | Author |
|---|---|---|
| National Institutes of Health | GM-039334 | Barbara Imperiali |
| National Institutes of Health | GM-131627 | Barbara Imperiali |
| National Institutes of Health | GM-134576 | Greg J Dodge |

The funders had no role in study design, data collection and interpretation, or the decision to submit the work for publication.

### Author contributions

Greg J Dodge, Conceptualization, Data curation, Formal analysis, Funding acquisition, Validation, Investigation, Visualization, Methodology, Writing – original draft, Writing – review and editing; Alyssa J Anderson, Formal analysis, Investigation, Methodology, Writing – original draft, Writing – review and

editing; Yi He, Rosa Viner, Formal analysis, Validation, Visualization, Methodology, Writing – review and editing; Weijing Liu, Validation, Investigation, Visualization, Methodology, Writing – review and editing; Barbara Imperiali, Conceptualization, Supervision, Funding acquisition, Project administration, Writing – review and editing

**Author ORCIDs**
Greg J Dodge (ORCID) https://orcid.org/0000-0002-6555-8350
Barbara Imperiali (ORCID) http://orcid.org/0000-0002-5749-7869

Reviewer #1 (Public Review): https://doi.org/10.7554/eLife.91125.2.sa1
Reviewer #2 (Public Review): https://doi.org/10.7554/eLife.91125.2.sa2
Author Response https://doi.org/10.7554/eLife.91125.2.sa3

## Additional files

**Supplementary files**
• Supplementary file 1. Additional tables relating to the initial crosslinking screen, EM structure data and processing, and NanoDSF ligand screening. (A) Screening chemical crosslinkers for the detection of *S. enterica* WbaP oligomers. *S. enterica* WbaP in styrene-maleic acid liponanoparticle (SMALP) was reacted with a panel of lysine-reactive crosslinkers with variable chemistries, solubilities, and lengths. Samples were analyzed by Western blot to detect the presence of crosslinked oligomers. Crosslinking efficiency was annotated as high (>30% total protein crosslinked to dimer), moderate (<30% total protein crosslinked to dimer), or none (no protein crosslinked to dimer). (B). Cryo-EM data collection, refinement, and validation statistics. (C) NanoDSF nucleotide ligand screen for soluble WbaP DUF truncation.

• MDAR checklist

**Data availability**
CryoEM density has been deposited in the Electron Microscopy Data Bank with the accession code EMD-41042. The corresponding coordinates have been deposited in the Protein Data Bank with the accession code 8T53. Mass spectrometry data, full-length WbaP dimer model generated by Alpha-Fold, and Foldseek output are available on Mendeley: https://doi.org/10.17632/zddxv2k83x.1.

The following datasets were generated:

| Author(s) | Year | Dataset title | Dataset URL | Database and Identifier |
|---|---|---|---|---|
| Dodge GJ, Imperiali B | 2024 | WbaP Structure associated data | http://dx.doi.org/10.17632/zddxv2k83x.1 | Mendeley Data, 10.17632/zddxv2k83x.1 |
| Dodge GJ, Imperiali B | 2024 | S. enterica WbaP in a styrene maleic acid liponanoparticle | https://www.ebi.ac.uk/emdb/EMD-41042 | Electron Microscopy Data Bank, EMD-41042 |
| Dodge GJ, Imperiali B | 2024 | S. enterica WbaP in a styrene maleic acid liponanoparticle | https://www.rcsb.org/structure/8T53 | RCSB Protein Data Bank, 8T53 |

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
