## [Editor Report · eLife assessment]

This **valuable** manuscript provides **solid** methodologies for utilizing SMALP nanodisks for oligomer characterization. The authors present a platform for capturing and studying native membrane protein oligomerization and subsequent cryoEM analysis. The specific application of the method to WbaP, a membrane-bound phosphoglycosyl transferase, adds to our understanding of glycoconjugate production in bacteria. This manuscript would be of interest to those focusing on native membrane protein studies and antimicrobial resistance.

---

## [Referee Report · Reviewer #1 (Public Review)]

Summary:

The authors characterize *S. enterica* WbaP biochemically and structurally. The enzyme catalyzes the initial step in O antigen biosynthesis by transferring a phospho-galactosyl unit from UDP-galactose to undecaprenyl-phosphate. This initial primer is then extended by other glycosyltransferases to form the O antigen repeat unit.

To preserve the biologically functional unit of WbaP, the authors chose a 'detergent-free' purification method based on membrane extraction using SMALP polymers. The obtained material was characterized biochemically and by single-particle cryo-electron microscopy.

Strengths:

The authors were able to isolate WbaP in a catalytically active and oligomeric form and determined a low-resolution cryo-EM structure of the dimeric complex. Using a disulfide cross-linking approach and other biophysical methods, the authors validated an AlphaFold predicted WbaP model used to interpret the experimental cryo-EM map.

Weaknesses:

The rationale for using SMALP to extract WbaP from the membrane was to 'preserve' the native lipid bilayer surrounding the protein. However, the physical properties of the lipids co-purifying with the protein are unclear. The volume of the EM map assigned to the SMALP polymers suggests a more micellar character.

Overall, the obtained cryo-EM map appears to be at fairly low resolution. Based on Figure 6, individual helices are not resolved, suggesting an overall resolution significantly below the stated 4.1 Å. Thus, the presented structure is the one of an AlphaFold WbaP model.

I believe the UMP titration analysis could be improved. The authors assume that a 'domain of unknown function (DUF)' binds UMP and regulates the enzyme's activity. UMP, a reaction product of WbaP, may also inhibit the enzyme competitively. Therefore, deleting the DUF for the UMP inhibition studies could help with data interpretation.

---

## [Referee Report · Reviewer #2 (Public Review)]

Summary:

The authors focused on delivering a comprehensive structural characterization of WbaP, a membrane-bound phosphoglycosyl transferase from *Salmonella* that is instrumental in bacterial glycoconjugate synthesis. Notably, the authors employed SMALP-200, an amphipathic copolymer, to extract WbaP in the form of native lipid bilayer nanodiscs. They then determined its oligomerization state through cross-linking and procured higher-resolution structural data via cryo-electron microscopy (cryo-EM). While the authors successfully characterized WbaP in a native-like lipid bilayer setting, and their findings support this, the paper's claim of introducing a novel methodology is not robust. The real contribution of this work lies in the newfound insights about WbaP's structure.

Strengths:

The manuscript provides novel insights into WbaP's structure and oligomerization state, highlighting potentially significant interactions. The methodologies employed represent state-of-the-art practices in the field. Most of the drawn conclusions are well-supported by either experimental or computational data, with a few exceptions noted below.

Weaknesses:

• Organization: The manuscript's organization lacks clarity. The authors seem to describe their processes in the sequence they occurred rather than a logical flow, leading to potential confusion. For instance, the authors delve into a series of inconclusive experiments to determine the oligomerization state of WbaP, utilizing techniques like SEC, SEC-MALS, mass photometry, and mass spectrometry. They then transition to cryo-EM but subsequently return to address the oligomerization issue, which they conclusively resolve using cross-linking experiments. Following this, they shift their focus to interpreting and discussing the structural features obtained from the cryo-EM data.

• Ambiguous and incorrect statements: There are instances of vague and at times inaccurate statements. Using more precise terminology like "native nanodiscs" or "lipid bilayer nanodiscs" would enhance clarity compared to the term "liponanoparticles." The claim on page 8 concerning the refractive index increment of SMA polymers needs rectification. The real reason why SEC-MALS cannot provide absolute particle masses in this case is that using two independent concentration detectors (typically, absorbance and refractive index), the decomposition of elution profiles is necessarily limited to two chemical species of a known molar or specific absorbance and refractive index. Thus, it is clear that nanodiscs containing a protein, a polymer, and a chemically undefined mixture of native lipids cannot be analyzed by this technique.

• Overstating of technical aspects: The technical aspects seem overstated. While the extraction of membrane proteins into native lipid bilayer nanodiscs and their characterization by cross-linking and cryo-EM are standard (and were published before by the same authors in ref. 29), the authors appear to promote them as groundbreaking. The statement that this study presents a novel, universal strategy and toolkit for examining small membrane proteins within liponanoparticles seems overstated, especially given the previous existence of similar methods.

---

## [Author Response]

**Public Reviews:**

**Reviewer #1 (Public Review):**
Summary:The authors characterize *S. enterica* WbaP biochemically and structurally. The enzyme catalyzes the initial step in O antigen biosynthesis by transferring a phospho-galactosyl unit from UDP-galactose to undecaprenyl-phosphate. This initial primer is then extended by other glycosyltransferases to form the O antigen repeat unit.To preserve the biologically functional unit of WbaP, the authors chose a 'detergent-free' purification method based on membrane extraction using SMALP polymers. The obtained material was characterized biochemically and by single-particle cryo-electron microscopy.Strengths:The authors were able to isolate WbaP in a catalytically active and oligomeric form and determined a low-resolution cryo-EM structure of the dimeric complex. Using a disulfide cross-linking approach and other biophysical methods, the authors validated an AlphaFold predicted WbaP model used to interpret the experimental cryo-EM map.Weaknesses:The rationale for using SMALP to extract WbaP from the membrane was to 'preserve' the native lipid bilayer surrounding the protein. However, the physical properties of the lipids co-purifying with the protein are unclear. The volume of the EM map assigned to the SMALP polymers suggests a more micellar character.Overall, the obtained cryo-EM map appears to be at fairly low resolution. Based on Figure 6, individual helices are not resolved, suggesting an overall resolution significantly below the stated 4.1 Å. Thus, the presented structure is the one of an AlphaFold WbaP model.I believe the UMP titration analysis could be improved. The authors assume that a 'domain of unknown function (DUF)' binds UMP and regulates the enzyme's activity. UMP, a reaction product of WbaP, may also inhibit the enzyme competitively. Therefore, deleting the DUF for the UMP inhibition studies could help with data interpretation.

We appreciate the reviewer’s careful analysis of our manuscript, and their attention to detail regarding the structural data. In a revised version of this manuscript, we will modify the discussion section to include a brief section focused on the liponanoparticle itself, comparing to other experimental structures in SMALP. Investigating the lipid microenvironment in SMALPs around both Lg- and Sm-PGTs is of great interest to our group. We have published initial data related to PglC from Campylobacter, but a systematic analysis of co-purified lipids from the growing number of SMALP-solubilized PGTs is an exciting future direction for this project. Expression and analysis of truncated constructs containing the catalytic domain of Lg-PGTs (including WbaP) has been attempted in our laboratory, with no success. This limits our ability to decouple DUF-mediated modulation of activity from interactions in the catalytic domain. Efforts to address this challenge are underway but will be the focus of future publications.Regarding the overall resolution – for transparency - we will add a new figure that shows the local resolution throughout the experimental map.

**Reviewer #2 (Public Review):**
Summary:The authors focused on delivering a comprehensive structural characterization of WbaP, a membrane-bound phosphoglycosyl transferase from *Salmonella* that is instrumental in bacterial glycoconjugate synthesis. Notably, the authors employed SMALP-200, an amphipathic copolymer, to extract WbaP in the form of native lipid bilayer nanodiscs. They then determined its oligomerization state through cross-linking and procured higher-resolution structural data via cryo-electron microscopy (cryo-EM). While the authors successfully characterized WbaP in a native-like lipid bilayer setting, and their findings support this, the paper's claim of introducing a novel methodology is not robust. The real contribution of this work lies in the newfound insights about WbaP's structure.Strengths:The manuscript provides novel insights into WbaP's structure and oligomerization state, highlighting potentially significant interactions. The methodologies employed represent state-of-the-art practices in the field. Most of the drawn conclusions are well-supported by either experimental or computational data, with a few exceptions noted below.Weaknesses:• Organization: The manuscript's organization lacks clarity. The authors seem to describe their processes in the sequence they occurred rather than a logical flow, leading to potential confusion. For instance, the authors delve into a series of inconclusive experiments to determine the oligomerization state of WbaP, utilizing techniques like SEC, SEC-MALS, mass photometry, and mass spectrometry. They then transition to cryo-EM but subsequently return to address the oligomerization issue, which they conclusively resolve using cross-linking experiments. Following this, they shift their focus to interpreting and discussing the structural features obtained from the cryo-EM data.• Ambiguous and incorrect statements: There are instances of vague and at times inaccurate statements. Using more precise terminology like "native nanodiscs" or "lipid bilayer nanodiscs" would enhance clarity compared to the term "liponanoparticles." The claim on page 8 concerning the refractive index increment of SMA polymers needs rectification. The real reason why SEC-MALS cannot provide absolute particle masses in this case is that using two independent concentration detectors (typically, absorbance and refractive index), the decomposition of elution profiles is necessarily limited to two chemical species of a known molar or specific absorbance and refractive index. Thus, it is clear that nanodiscs containing a protein, a polymer, and a chemically undefined mixture of native lipids cannot be analyzed by this technique.• Overstating of technical aspects: The technical aspects seem overstated. While the extraction of membrane proteins into native lipid bilayer nanodiscs and their characterization by cross-linking and cryo-EM are standard (and were published before by the same authors in ref. 29), the authors appear to promote them as groundbreaking. The statement that this study presents a novel, universal strategy and toolkit for examining small membrane proteins within liponanoparticles seems overstated, especially given the previous existence of similar methods.

We appreciate the reviewer’s careful consideration of the steps that were taken and how they were presented. However, we need to reinforce that although the initial biophysical experiments do not provide the exact oligomeric state of the WbaP, they provide important new data. Together these data support that the intact liponanoparticle is large enough to accommodate a higher order oligomerization state along with native lipids and stabilizing SMA polymer – this was not known at the outset and led to Fig 2D showing the first demonstration of dimer that was then validated via XLMS and disulfide crosslinking. The process was logical and essential to this work. We recognize the reviewer’s point on the SEC-MALs experiment and will adjust the text accordingly.

We sought to distinguish the stabilization method used here from canonical MSP nanodiscs by using the term styrene maleic acid liponanoparticle (SMALP). The term SMALP is widely used in literature utilizing this technology, thus the use of other terms may lead to confusion.

Our manuscript in PExpPur was focused on enabling expression of sufficient quality and quantity for sophisticated downstream biophysical applications – that MS was intended to be enabling to the greater membrane protein community and is highly recognized and appreciated in “its own right.” This work presents the first in class structure of the large monoPGTs. Further only a single structure of the PGT domain itself has been solved and appears as an experimental structure in the PDB (also from our group) addressing the enigmatic additional domains and potential physiological relevance. It is also noteworthy that the Lg-monoPGTs dominate the superfamily. This is also the first time that any protein in SMALP has been characterized using direct mass technology, which provided the most accurate mass determination of the intact liponanoparticle/protein complex.